# Soil water content drives the spatiotemporal the distribution and community assembly of soil ciliates in the Nianchu River Basin, Qinghai-Tibet Plateau, China

**Shiying Zhu**[1], **Qian Huang**[1], **Tianshun Li**[1], **Mingyan Li**[1], **Qing Yang**[2,3], **Xiaodong Li**[2,3], **Alan Warren**[4], **Bu Pu** [1] *

**1** Department of Life Sciences, Plateau Zoology Laboratory, School of Ecology and Environment, Tibet University, Lhasa, China, **2** Laboratory of Wetland and Watershed Ecosystems of Tibetan Plateau, School of Ecology and Environment, Tibet University, Lhasa, China, **3** Center for Carbon Neutrality in the Third Pole of the Earth, Tibet University, Lhasa, China, **4** Department of Life Sciences, Natural History Museum, London, United Kingdom

* purbu@utibet.edu.cn

**Data Availability Statement:** All relevant data are within the manuscript and its Supporting Information files.

## Abstract

Ciliated protozoa (ciliates) are an ecologically important group of microeukaryotes that play roles in the flow of energy and nutrients in aquatic and terrestrial ecosystems. The community distribution and diversity of soil ciliates in the Nianchu River Basin were investigated by sampling four major habitats, i.e., grassland, farmland, wetland and sea buckthorn forest during May, August and October 2020. Cultivation identification and enumeration of soil ciliates were performed by the non-submerged culture method, in vivo observations and protargol silver staining, and direct counting methods, respectively. A total of 199 species were identified representing, 89 genera, 67 families, 31 orders and 11 classes. Haptorida was the dominant group with 35 species, accounting for 17.59% of the total. The results showed that the α and β diversity indices of soil ciliate communities in the Nianchu River Basin varied significantly in spatial distribution, but not in temporal distribution. Mantel test showed that soil water content, total nitrogen and organic matter were significantly correlated with soil ciliates. Soil water content was the main environmental factor driving the spatial distribution of soil ciliates. Co-occurrence network analysis showed that soil ciliate species in the Nianchu River Basin depend on each other in the relationship of solidarity and cooperation or ecological complementarity. Thus maintaining or enhancing the diversity and stability of the community. Community assembly shows that randomness process was an important ecological process driving soil ciliate community construction in the Nianchu River Basin.

## Introduction

As one of the most complex ecosystems in nature, soil is the link between the hydrosphere, atmosphere, lithosphere and biosphere [1], and plays an important role in maintaining the

**Funding:** Comprehensive Scientific Investigation Project of Biodiversity Survey and Maintenance Mechanism Evaluation in the "One River and Four Rivers" Basin (Zangcai Science Education Index (2021) No.1 and Zangcai Education Index (2019) 01); Supported by "High-level Talents Cultivation Program" for Graduate Students of Tibet University, No.2020-GSP-S042.

**Competing interests:** The authors declare no competing interests.

ecological functions and services of terrestrial ecosystems [2]. Soil is a substrate with highly heterogeneous spatial structure and complex chemical composition thus providing a large number of habitats for soil microbial communities [3]. As an integral part of the soil microbial community, ciliates play a key role in the food chain by linking bacteria with higher trophic levels [4], and by regulating ecological processes such as energy flow, material circulation, and information exchange [5,6]. The abundance of soil ciliates is highly variable but can be as high as 10–500 cells/g [7]. Soil ciliates generally live in soil particles, soil pore water or litter water membrane on soil surfaces [8]. They play an important role in soil ecosystems. Ciliates are regarded as reliable indicators of soil ecology due to their wide distribution, high species diversity, small size, rapid community succession, and rapid response to small changes in the external environment [9] and thus can be used to monitor soil environmental quality [10]. Most studies on soil ciliates have focused on morphological description and community characteristics analysis [11–13]. The community assembly process, i.e., that by which species settle and interact to establish and sustain local communities through successive repeated migrations from regional species banks and its driving factors have been rarely reported. The Qinghai-Tibet Plateau is variously referred to as the "water tower of Asia", the "third pole of the Earth" and the "roof of the world". It has important functions of water conservation, soil conservation, windbreak and sand fixation, carbon fixation and biodiversity protection [14]. It is an important ecological security barrier area for Asia and for China in particular and is a global hotspot for biodiversity conservation. The Nianchu River Basin is located in the southeast of the Qinghai-Tibet Plateau, which contains rich biodiversity resources. Various plateau ecosystems distributed within the basin play important roles in the construction of the ecological barrier of the Qinghai-Tibet Plateau [15]. These ecosystems include wetlands, grasslands, farmland and forests. Wetlands contribute to various aspects of the water cycle including water conservation, maintaining and/or improving water quality, and regulating runoff, [16]. Grasslands are among the most important carbon sinks of the terrestrial ecosystem. Forests have important social and natural economic value. Farmland supports energy conversion and food production according to the needs of human society; crop-centered farmland, for example, utilize the interrelationships between the terrestrial environments and biological populations by optimizing ecological structure and efficient ecological function [17]. In this paper, the characteristics, driving factors (factors affecting soil ciliate community) and community assembly mechanism of soil ciliate communities were investigated in various habitats in the Nianchu River Basin. The main aims of this study were to: (1) monitor the changes of soil environment in the basin through the dynamic changes of soil ciliate community; (2) provide baseline data for the environmental protection of the Nianchu River Basin and as a scientific basis for the sustainable development and biodiversity (biodiversity is the basis, goal and means of sustainable development) conservation in the Qinghai-Tibet Plateau.

## Materials and methods

### Study area profile

The Nianchu River Basin is located on the right bank of the middle Yarlung Zangbo River. Water in the basin feeds the Nianchu River which originates from the northern foothills of the middle Himalayan Mountains and flows into the Yarlung Zangbo River from the southwest to the north. The basin is located between 28°10' N -29°20' N and 88°35' E -90°15' E. The basin covers an area of about 11130km$^2$, with a river length of 217km and a total drop of 1322 meters. The average slope drop is 6.1% [16,18]. The basin is within the warm, semi-arid plateau monsoon climate region, with climate characteristics of low temperature, large sunlight radiation, dry air, little precipitation, and large temperature differences between morning and

evening. The dry and wet seasons are distinct. The rainy season is from June to September, and the annual precipitation is 429 mm [19,20]. Common plants in the basin include *Hippophae* sp, *Androsace tapeta*, and *Potentilla anserina*. Common animals include *Procapra picticaudata*, *Equus kiang*, and *Ochotona curzoniae*.

## Sampling method

Based on the distribution characteristics of habitats in the Nianchu River Basin, 19 sampling sites, each 20 m x 20 m, were set up according to the geographical distance between habitats (Fig 1). There were four sampling sites on grassland and five each on farmland, wetland, and sea buckthorn forest (Table 1). Three samples were collected in May (spring), August (summer), and October (Autumn) during 2020. GPS (model: MAP63lcsx) was used to record the altitude, latitude and longitude of each sample site. Soil surface temperature (ST) was measured with a soil temperature gun (Model: XTY0087). Five quadrants (20cmx20cm) were selected by the "five-point sampling method of plum blossom", and 0-10cm soil samples were collected using a soil collection device (QTZ, Sanyu) after removing the litter from the soil surface. Therefore, a total of 285(19×5×3) soil samples were obtained. The collected samples were sealed and brought back to the laboratory for processing. Part of each sample was used for the

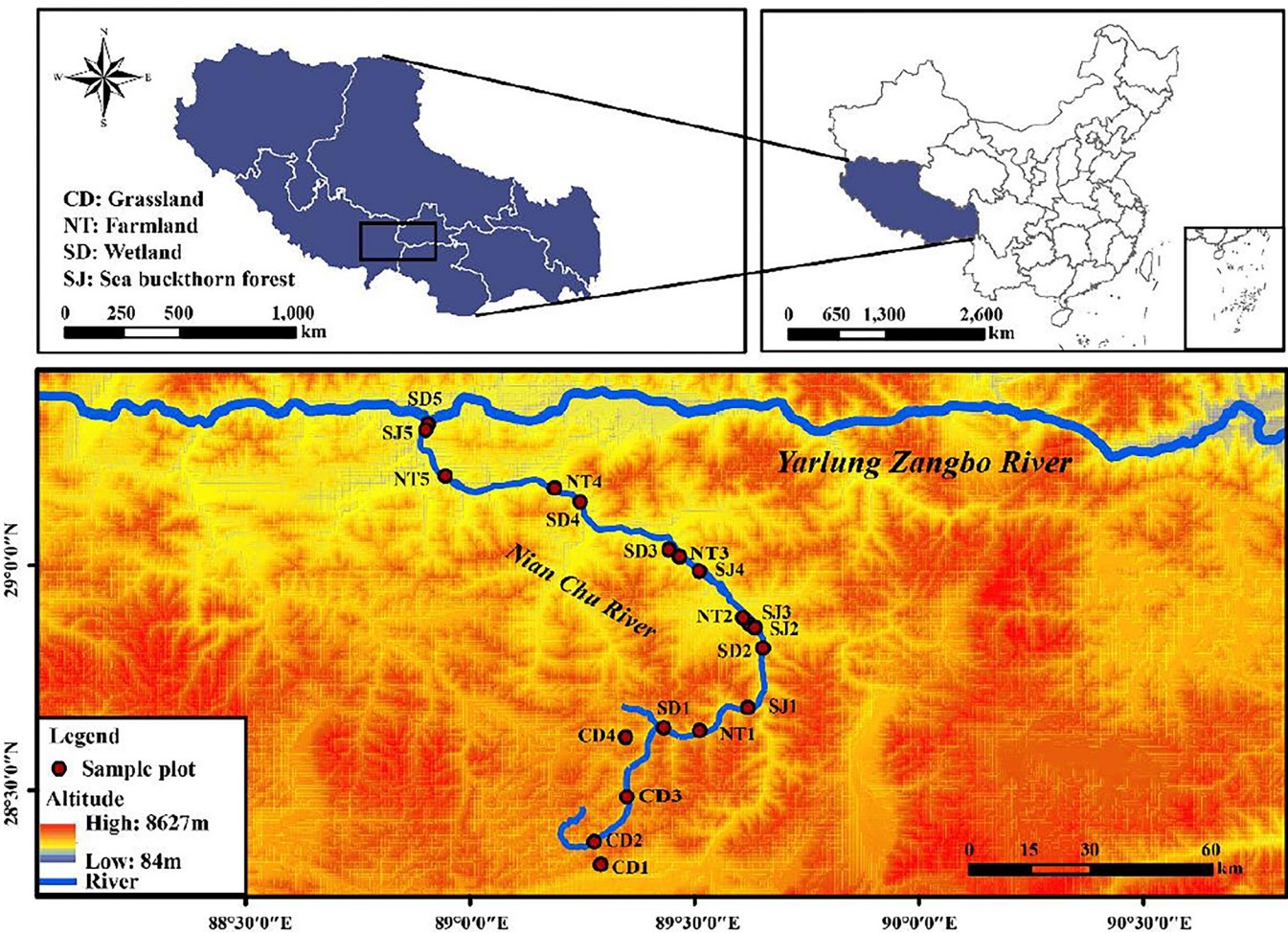

**Fig 1. The sampling site of Nianchu River basin.**

**Table 1. Geographical data on ecosystems in the Nianchu River Basin.**

| Sites | Altitude/m | Latitude and Longitude |
|---|---|---|
| Grassland | 4802–4530 | N:28°20′04.58″~28°37′03.34″E:89°17′31.53″~89°20′51.54″ |
| Farmland | 4318–3844 | N:28°37′54.47″~29°11′54.28″E:89°30′42.63″~88°56′43.98″ |
| Wetland | 4394–3849 | N:28°38′19.39″~29°08′27.79″E:89°25′57.23″~89°14′45.58″ |
| Sea buckthorn forest | 4165–3833 | N:28°41′00.96″~29°18′02.96″E:89°37′10.91″~88°54′06,95″ |

determination of physical and chemical factors. The rest which was dried by spreading evenly on kraft paper and was covered with papyrus paper to prevent ciliate cysts floating in the air from affecting the subsequent experimental results. After being naturally air-dried in a dry and cool place (about 30 days), the samples were used for ciliate cultivation and identification.

## Culture and identification of soil ciliates

Using an electronic balance, 50g of air-dried soil was placed in a 15 cm diameter Petri dish. The natural conditions during sampling were simulated by using a light incubator. The soil ciliates were cultured by the "non-submerged culture method" whereby soil leachate solution was added to the culture dish to make the soil sufficiently moist but not submerged [21]. After 24 hours of culture, samples were examined by light microscopy (Leica DM500 microscope) to observe specimens *in vivo*. Soil leachate solution was added regularly to keep the soil samples in a moist but non-submerged state. The protargol staining method [22] was used to reveal the infraciliature thus allowing definitive species identification using published keys and guides [23–29]. Classification follows Lynn [30].

## Determination of physicochemical factors

The soil water content (SWC) was determined by the wet-dry specific gravity method. Other soil physical and chemical factors were determined by Tibet Boyuan Environmental Testing Co., LTD. Soil pH was determined using a pHS-3C pH meter. Soil total nitrogen (TN) and soil organic matter (SOM) were determined by acid buret. Rapid available potassium (RAK) was determined by flame photometry. Available phosphorus (AP) was determined by sodium bicarbonate extraction and molybdenum-antimony resistance colorimetry [31].

## Keystone species

The importance of nodes is measured by the inter-module connectivity ($Z_i$) and inter-module connectivity ($P_i$). When $Z_i < 2.5$ and $P_i < 0.62$, the network node is considered unimportant. When $Z_i < 2.5$ and $P_i > 0.62$, the node is considered to be a connector. When $Z_i > 2.5$ and $P_i < 0.62$, the node is considered to be a module hub. When $Z_i > 2.5$ and $P_i > 0.62$, the node is considered to be a network hub. Connectors, module hubs and network hubs are all considered to be keystone species in the construction of networks and may play important roles in the maintenance of network structure [32].

$$Z_i = \frac{K_i - \bar{K}_{si}}{\sigma K_{si}} \tag{1}$$

$$P_i = 1 - \sum_{s=1}^{N_m} \left(\frac{K_{si}}{K_i}\right)^2 \tag{2}$$

Where, $Z_i$ is the connectivity degree of node i module, $K_i$ indicates the connectivity of

module i of node, $\bar{K}_{si}$ is the average value of $K_i$ of all nodes in module s where node i resides, $\sigma K_{si}$ is the standard deviation of connectivity within the module of all nodes in module s where node i resides, $P_i$ is the participation coefficient of node i, Nm is the number of modular, s indicates the module s, and $K_{is}$ is the connectivity of node i in each module

### Neutral community model

The neutral model quantifies the importance of stochastic processes. $R^2$ represents the degree of fit of the neutral model as a whole. N describes the size of the meta-community, m quantifies the mobility at the community level, and Nm quantifies the diffusion between communities [33].

### C/P quotient

Ciliate classes that are commonly represented in soils ciliate communities include Colpoldea, Spirotrichea, Heterotrichea, and Armophorea. The C/P quotient is the proportion of the usually r-selected Colpodea species and the usually k-selected Spirotrichea, Heterotrichea, and Armophorea species and is an important measure of habitat conditions. When $C/P \leq 1$, the environmental condition in the soil habitat is relatively good. When $C/P > 1$, the environmental condition of the soil habitat is relatively poor [13,34].

### Data processing and analysis

The Shannon-Wiener Diversity Index [35,36], Margalef Richness Index, Non-metric multidimensional scaling, Co-occurrence network, Key Species, Mantel test and community construction were used R 4.0.5 to calculated and diagramed. The R packages used include vegan [37], psych [38], microeco [39], igraph [40], rlang [41], RColorBrewer [42], ggplot2 [43]. The Shannon-Weiner diversity index and Margalef richness index were calculated as follows:

Shannon-Wiener diversity index:

$$H = -\sum_{i=1}^{s} \frac{n_i}{N} ln\left(\frac{n_i}{N}\right) \tag{3}$$

Margalef richness index:

$$D = \frac{(S-1)}{lnN} \tag{4}$$

Where, H is the species diversity index, D is the richness index, $n_i$ is the number of individuals of class i, N is the total number of individuals, and S is the number of all groups.

## Results

### Species composition

A total of 199 species of soil ciliates were identified the four ecosystems that were sampled in the Nianchu River Basin. These represented 89 genera, 67 families, 31 orders and 11 classes (Table 2). Haptorida was the dominant group with 35 species, accounting for 17.59% of the total. Followed by Sessilida with 19 species, accounting for 9.55% of the total. Protostomatida, Cyclotrichiida, Bryophryida, Plagiopylida, Odontostomatida, Strombidiida and Endogenida were comparatively rare group with seven species in total, accounting for 3.52% of the total. Spatially, soil ciliates in the grassland ecosystem were dominated by members of the class Oligohymenophorea.The majority of soil ciliates in the farmland, wetland and sea buckthorn forest ecosystems were members of the class Spirotrichea (Fig 2A). Forty-six ciliates species were common for the four ecosystems. The sea buckthorn forest had the fewest ecosystem-specific

**Table 2. Composition of soil ciliate communities in the Nianchu River Basin.**

| Class | Orders | Families | Genus | Species | Percent |
|---|---|---|---|---|---|
| Oligohymenophorea | Pleuronematida | Pleuronematidae | 1 | 1 | 0.50% |
| | | Cyclidiidae | 2 | 10 | 5.03% |
| | Sessilida | Scyphidiidae | 1 | 1 | 0.50% |
| | | Operculariidae | 1 | 2 | 1.01% |
| | | Epistylididae | 1 | 3 | 1.51% |
| | | Vorticellidae | 3 | 13 | 6.53% |
| | Ophryoglenida | Ophryoglenidae | 1 | 2 | 1.01% |
| | Philasterida | Cohnilembidae | 1 | 2 | 1.01% |
| | | Uronematidae | 1 | 1 | 0.50% |
| | | Loxocephalidae | 1 | 2 | 1.01% |
| | | Cinetochilidae | 2 | 4 | 2.01% |
| | Tetrahymenida | Glaucomidae | 3 | 6 | 3.02% |
| | | Tetrahymenidae | 1 | 1 | 0.50% |
| | | Turaniellidae | 1 | 2 | 1.01% |
| | Peniculida | Lembadionidae | 1 | 1 | 0.50% |
| | | Frontoniidae | 1 | 2 | 1.01% |
| | | Parameciidae | 1 | 1 | 0.50% |
| Karyorelictea | Loxodida | Loxodidae | 1 | 2 | 1.01% |
| | Protostomatida | Trachelocercidea | 1 | 1 | 0.50% |
| Nassophorea | Synhymeniida | Scaphidiodontidae | 1 | 2 | 1.01% |
| | Nassulida | Nassulidae | 1 | 3 | 1.51% |
| | | Furgasoniidae | 1 | 2 | 1.01% |
| | Microthoracida | Leptopharyngidae | 1 | 2 | 1.01% |
| Litostomatea | Pleurostomatida | Amphileptidae | 1 | 1 | 0.50% |
| | | Litonotidae | 2 | 7 | 3.52% |
| | Haptorida | Spathidiidae | 2 | 16 | 8.04% |
| | | Trachelophyllidae | 2 | 5 | 2.51% |
| | | Actinobolinidae | 1 | 1 | 0.50% |
| | | Enchelyidae | 2 | 2 | 1.01% |
| | | Tracheliidae | 3 | 7 | 3.52% |
| | | Lacrymariidae | 1 | 2 | 1.01% |
| | | Didiniidae | 1 | 1 | 0.50% |
| | | Acropisthiidae | 1 | 1 | 0.50% |
| | Entodiniomorphida | Ophryoscolecidae | 1 | 2 | 1.01% |
| | Cyclotrichiida | Mesodiniidae | 1 | 1 | 0.50% |
| Armophorea | Armophorida | Metopidae | 1 | 2 | 1.01% |
| Prostomatea | Prorodontida | Colepidae | 1 | 1 | 0.50% |
| | | Holophryidae | 1 | 4 | 2.01% |
| | | Prorodontidae | 2 | 5 | 2.51% |
| | | Urotrichidae | 1 | 4 | 2.01% |
| | | Plagiocampidae | 1 | 3 | 1.51% |
| | | Placidae | 1 | 1 | 0.50% |
| Colpodea | Cyrtolophosidida | Cyrtolophosididae | 1 | 4 | 2.01% |
| | | Platyophryidae | 1 | 2 | 1.01% |
| | Colpodida | Colpodidae | 2 | 6 | 3.02% |
| | | Grossglockneriidae | 1 | 1 | 0.50% |
| | Bryophryida | Bryophyidea | 1 | 1 | 0.50% |

*(Continued)*

**Table 2.** (Continued)

| Class | Orders | Families | Genus | Species | Percent |
|---|---|---|---|---|---|
| Plagiopylea | Plagiopylida | Plagiopylidae | 1 | 1 | 0.50% |
| | Odontostomatida | Epalxellidae | 1 | 1 | 0.50% |
| Spirotrichea | Strombidiida | Strombidiidae | 1 | 1 | 0.50% |
| | Stichotrichida | Spirofilidae | 1 | 2 | 1.01% |
| | | Keronidae | 1 | 1 | 0.50% |
| | Sporadotrichida | Halteriidae | 1 | 1 | 0.50% |
| | | Trachelostylidae | 1 | 1 | 0.50% |
| | | Oxytrichidae | 7 | 15 | 7.54% |
| | Urostylida | Urostylidae | 4 | 8 | 4.02% |
| | Euplotida | Euplotidae | 1 | 4 | 2.01% |
| | | Aspidiscidae | 1 | 1 | 0.50% |
| | Choreotrichida | Strobilidiidae | 1 | 2 | 1.01% |
| Phyllopharyngea | Endogenida | Acinetidae | 1 | 1 | 0.50% |
| | Exogenida | Podophryidae | 1 | 3 | 1.51% |
| | Chlamydodontida | Chilodonellidae | 1 | 6 | 3.02% |
| Heterotrichea | Heterotrichida | Blepharismidae | 1 | 2 | 1.01% |
| | | Stentoridae | 1 | 1 | 0.50% |
| | | Spirostomidae | 1 | 1 | 0.50% |
| | | Condylostomatidae | 1 | 1 | 0.50% |

species (four), whereas the wetland had the most cosystem-specific species (35) (Fig 2C). Spirotrichea dominated the ciliate communities in spring, summer and autumn (Fig 2B). The number of ecosystem-specific species was lowest in autumn (15) and highest in spring (20) (Fig 2D).

## C/P quotient

Spatially, grassland and farmland ecosystem had the largest C/P quotient (0.38) and sea buckthorn forest ecosystem had the smallest (0.19). Temporally, the largest C/P quotient (0.30) was recorded in autumn compared with 0.24 in spring The C/P quotients of soil ciliates in the Nianchu River Basin were invariably less than 1 (Table 3).

## α diversity of soil ciliate community

One-way Analysis of Variance (ANOVA) of the mean Shannon-Wiener diversity index showed that grassland (H = 2.83) > farmland (H = 2.77) > sea buckthorn forest (H = 2.56) > wetland (H = 2.51), and that there were significant differences ($P < 0.05$) between wetland, grassland and farmland (Fig 3). Furthermore, ANOVA of the mean Shannon-Wiener diversity index of showed that spring = summer (H = 2.74) > autumn (H = 2.47), and that there were significant differences between autumn and both spring and summer ($P < 0.05$). The average Margalef richness index for each ecosystem showed that wetland (M = 4.29) > grassland (M = 4.06) > farmland (M = 3.28) > sea buckthorn forest (M = 3.02), and there were no significant differences between grassland and each of the other ecosystems ($P > 0.05$). Temporally, average Margalef richness index showed that summer (M = 3.82) > spring (M = 3.65) > autumn (M = 3.27), and that there were no significant differences among seasons ($P > 0.05$). In general, the soil ciliate diversity indeces was differed spatially but not temporally in the Nianchu River Basin.

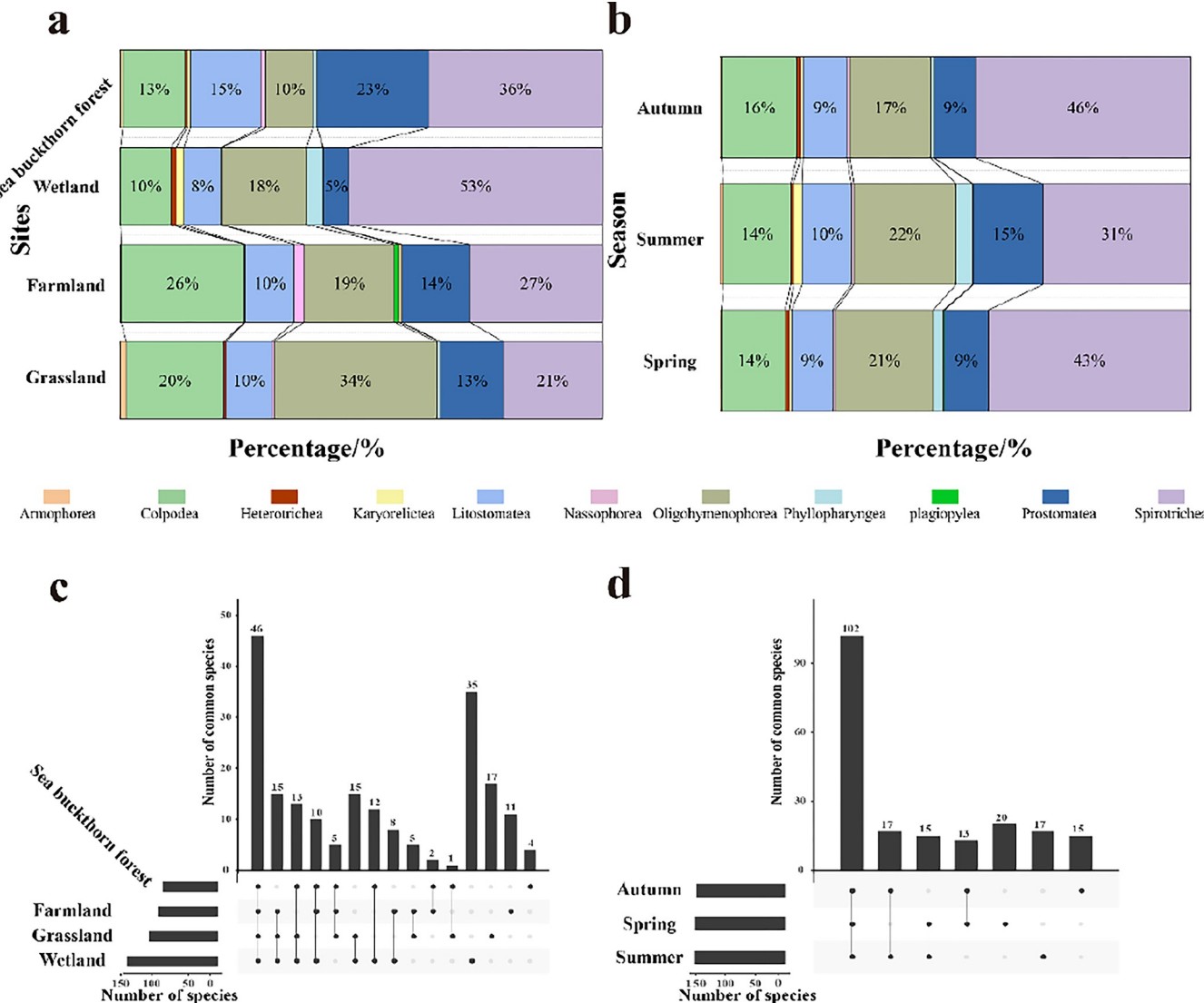

**Fig 2. Soil ciliate community composition with different spatial and temporal distributions in the Nianchu River Basin.** a: Community composition among different ecosystems based on class level; b: Interseasonal community composition based on class level; c: UpSet plot for ciliate communities from different ecosystems; d: UpSet plot for ciliate communities from different seasons.

## β diversity of soil ciliate community

There were differences in soil ciliate community composition based on different spatial and temporal distributions. The results of Non-metric multidimensional scaling based on Bray-Curtis dissimilarity revealed that the values of intersite and interseason pressure were all less than 0.3, indicating that it is a better ranking. Spatially, there were significant differences in soil ciliate community composition among the four ecosystems (P = 0.01). Particularly

**Table 3. Spatiotemporal distribution of C/P coefficient of soil ciliates in the Nianchu River Basin.**

| Sites | Grassland | Farmland | Wetland | Sea buckthorn forest | Spring | Summer | Autumn |
|---|---|---|---|---|---|---|---|
| C/P | 0.38 | 0.38 | 0.30 | 0.19 | 0.24 | 0.24 | 0.30 |

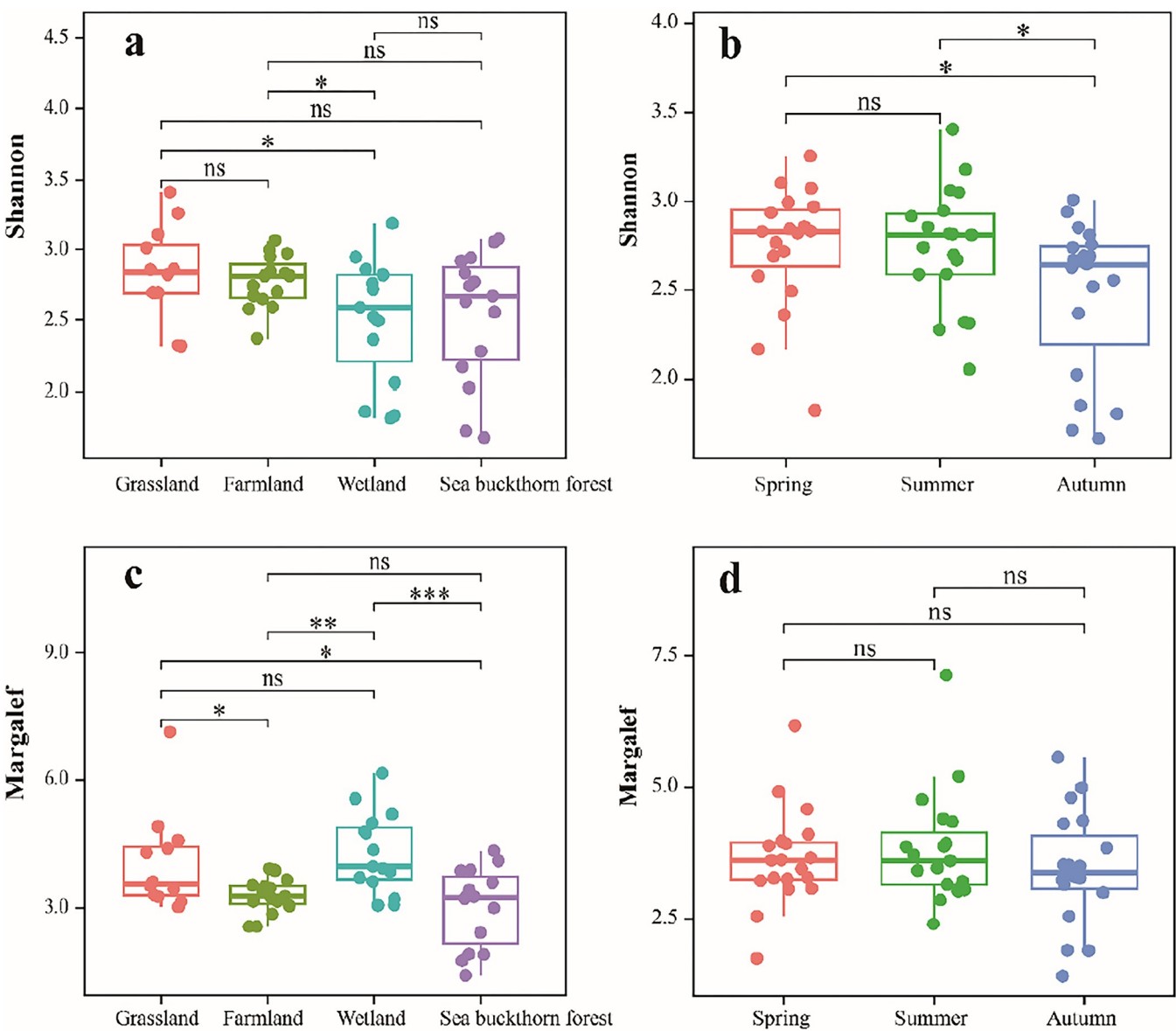

**Fig 3. Soil ciliate diversity with different temporal and spatial distributions in the Nianchu River Basin.** a: Shannon-Wiener diversity index among different ecosystems; b: Shannon-Wiener diversity index between seasons; c: Margalef richness index between different ecosystems; d: Margalef richness index between seasons. ns: P>0.05, *:0.01<P<0.05, **:0.001<P<0.01, ***:P<0.001.

between the wetland and the other three ecosystems (Fig 4A), There were no significant differences in soil ciliate community composition among spring, summer and autumn (P = 0.87). In general, the results of NMDS showed that soil ciliate communities in the Nianchu River Basin had significant spatial differences, but no significant seasonal differences.

## Soil ciliate co-occurrence network analysis and key species

Co-occurrence networks can help us understand the interrelationships between different species and how they coexist in ecosystems. By constructing co-occurrence networks of soil ciliate communities in each ecosystem and season (Fig 5A–5G), network level and node-level network topology characteristics among the networks were calculated (Fig 5O and Table 4).

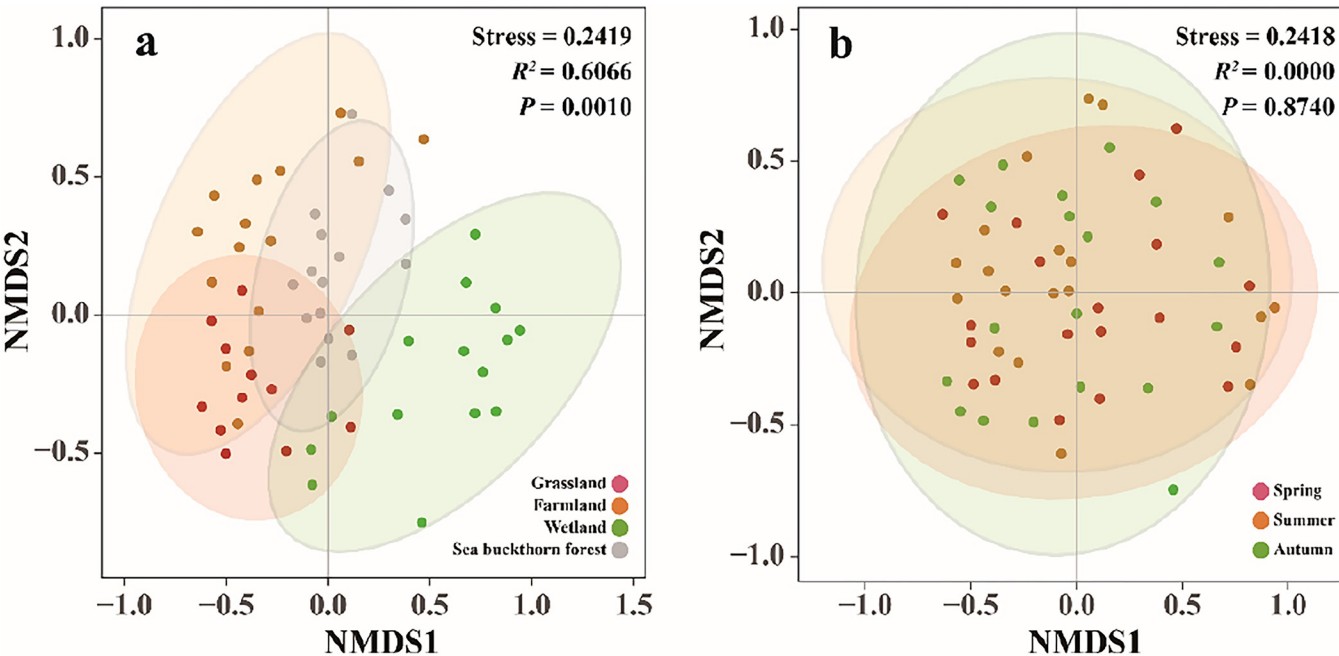

**Fig 4. Non-metric multidimensional scaling of soil ciliates with different spatial and temporal distributions in the Nianchu River Basin.** a: Analysis of NMDS in different ecosystems; b: Analysis of NMDS in different seasons.

Spatially, the number of nodes and edges of the wetland co-occurrence network was much larger than that of the other ecosystems, indicating that the ciliate community structure of the wetland was more complex. The average degree of the wetland cooccurrence network was the largest among the four ecosystems, indicating that the average interaction between soil ciliate communities was strong. The network density of alpine grassland was the highest among the four ecosystems, indicating that grassland soil ciliate species had closer interactions. The clustering coefficients of the wetland and sea buckthorn forest were the largest among the four ecosystems, indicating that the nodes of the wetland and sea buckthorn forest were more closely connected and had more information exchange. The mean path length of farmland was the largest among the four ecosystems, indicating that the distance between ciliate species in farmland soil was larger. Temporally, the number of edges and the clustering coefficients of the co-occurrence network in spring were larger than those in summer and autumn. The average degree of the autumn network was the largest among the three seasons, while the modularity of spring is the smallest. Summer had the largest mean path length of the three seasons. In general, soil ciliate species in the Nianchu River Basin were mainly cooperative in space and time, and there was little interspecific competition. No key species of soil ciliates exist between different ecosystems in the Nianchu River Basin (Fig 5H–5K). However, *Spathidium muscorum* in spring and *Cyclidium lanuginosum* in autumn were considered module hubs (Fig 5L–5N) and *Ophryoglena atra* and *Prorodon virides* in summer were considered connectors (Fig 5M). Therefore, these four species are considered key seasonal species, and their disappearance is likely to cause the collapse of the network structure between seasons.

## Analysis of environmental factors

Analysis of Variance (ANOVA) of environmental factors with different spatial and temporal distributions showed that there were no significant differences ($P > 0.05$) in AP, RAK and TN

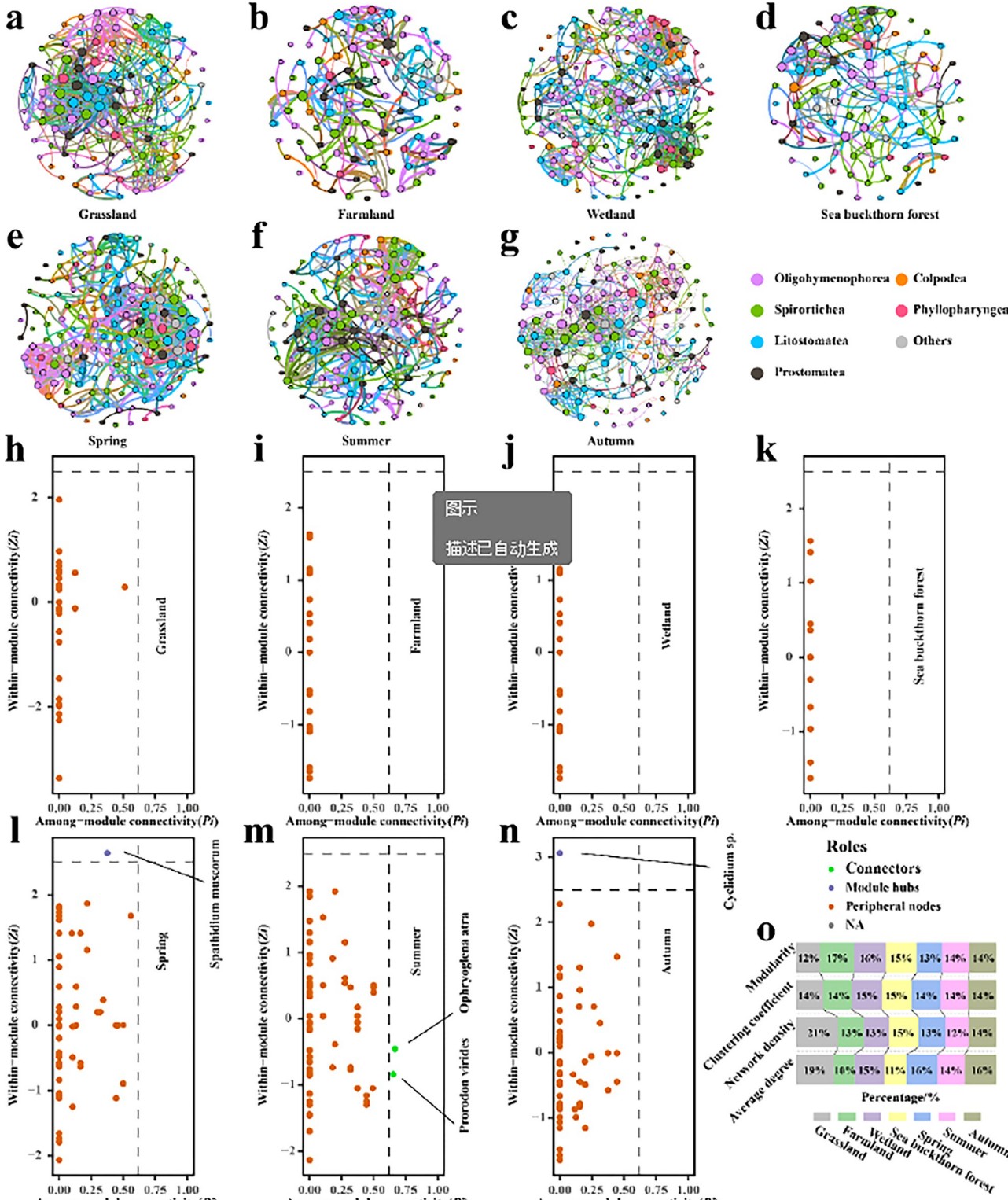

**Fig 5. Co-occurrence network and key species of soil ciliates with different temporal and spatial distributions in the Nianchu River Basin.** a: Soil ciliate co-occurrence network in grassland; b: ciliate co-occurrence network in farmland soil; c: Soil ciliate co-occurrence network in wetland; d: Soil ciliate co-occurrence network of Sea buckthorn forest; e: soil ciliate co-occurrence network in spring; f: soil ciliate co-occurrence network in summer; g: soil ciliate co-occurrence network in autumn; h: Keystone species of soil ciliates in grassland; i: Keystone species of soil ciliates in farmland; j: Keystone species of soil ciliates in wetland; k: Keystone species of soil ciliates in sea buckthorn forest; l: Keystone species of soil ciliates in spring; m: Keystone species of soil ciliates in summer; n: Keystone species of soil ciliates in autumn; o: Network topology.

**Table 4. Network topology.**

| Network topology indicators | Grassland | Farmland | Wetland | Sea buckthorn forest | Spring | Summer | Autumn |
|---|---|---|---|---|---|---|---|
| Number of nodes | 117 | 102 | 154 | 93 | 150 | 151 | 147 |
| Number of connections | 663 | 306 | 684 | 297 | 695 | 611 | 688 |
| Average degree | 11.3 | 6 | 8.88 | 6.39 | 9.27 | 8.09 | 9.36 |
| Modularity | 0.57 | 0.84 | 0.75 | 0.74 | 0.62 | 0.66 | 0.66 |
| Network density | 0.1 | 0.06 | 0.06 | 0.07 | 0.06 | 0.05 | 0.06 |
| Clustering coefficient | 0.67 | 0.7 | 0.72 | 0.72 | 0.7 | 0.68 | 0.69 |
| Mean path length | 3.38 | 5.07 | 4.34 | 4.52 | 4.14 | 4.37 | 4.25 |
| Positive correlation ratio(%) | 98.6 | 97.4 | 99.3 | 98.7 | 99.6 | 99.67 | 100 |
| Negative correlation ratio(%) | 1.36 | 2.61 | 0.73 | 1.35 | 0.43 | 0.33 | 0 |

among the four ecosystems (Fig 6). There were, however, significant differences in altitude, pH, SOM, ST and SWC among the four ecosystems (P < 0.05). There were no significant differences in altitude, SOM, ST, SWC and TN among seasons (P > 0.05), but there were significant differences in AP, pH and RAK among seasons (P < 0.05).

## Correlation between soil ciliates and environmental factors

In order to study the correlation between soil ciliate communities and environmental factors, the Mantel test was used to examine the community assembly process. Spatially, soil ciliate communities and SWC in farmland, and soil ciliate community and SWC plus TN in grassland had extremely significant correlations (P < 0.01) (Fig 7A). There were significant correlations between soil ciliate communities and SOM in grassland, between soil ciliate communities and TN and SOM in wetland, and between soil ciliate communities and SWC, TN and SOM in sea buckthorn forest (0.01 < P < 0.05). Seasonally (Fig 7B), there was no significant correlation between soil ciliate communities and environmental factors (P > 0.05). Linear fitting of SWC, TN and SOM based on Bray-Curtis dissimilarity (Fig 7C–7E) showed that with the increase of geographical distance, only SWC was significantly correlated with soil ciliate species (P = 0.0144). Therefore, the spatial distribution of soil ciliate species in the Nianchu River Basin was mainly driven by SWC.

## Construction of soil ciliate communities based on the neutral model

As shown in Fig 8, the frequency of occurrence of soil ciliate species in the Nianchu River Basin was mostly within the 95% confidence interval of the neutral model, indicating that the construction of soil ciliate communities was more influenced by random processes than by deterministic processes. The $R^2$ value of farmland was the largest, indicating that stochastic processes had the strongest driving effect on the accumulation of soil ciliate communities in farmland. The $R^2$ value was the smallest in autumn, indicating that stochastic processes had a weak driving effect on soil ciliate community assembly in autumn. The m value of the wetland was the smallest among the four ecosystems, indicating that soil ciliate communities in the wetland were the most restricted by random diffusion. The m value of farmland was the largest among the four ecosystems, indicating that soil ciliates in farmland were the least restricted by random diffusion. The maximum Nm value in the wetland indicated that the frequency of random diffusion had the strongest correlation with the relative abundance of species, while the minimum Nm value in summer indicated that the frequency of random diffusion had the weakest correlation with the relative abundance of species, in this ecosystem. In general,

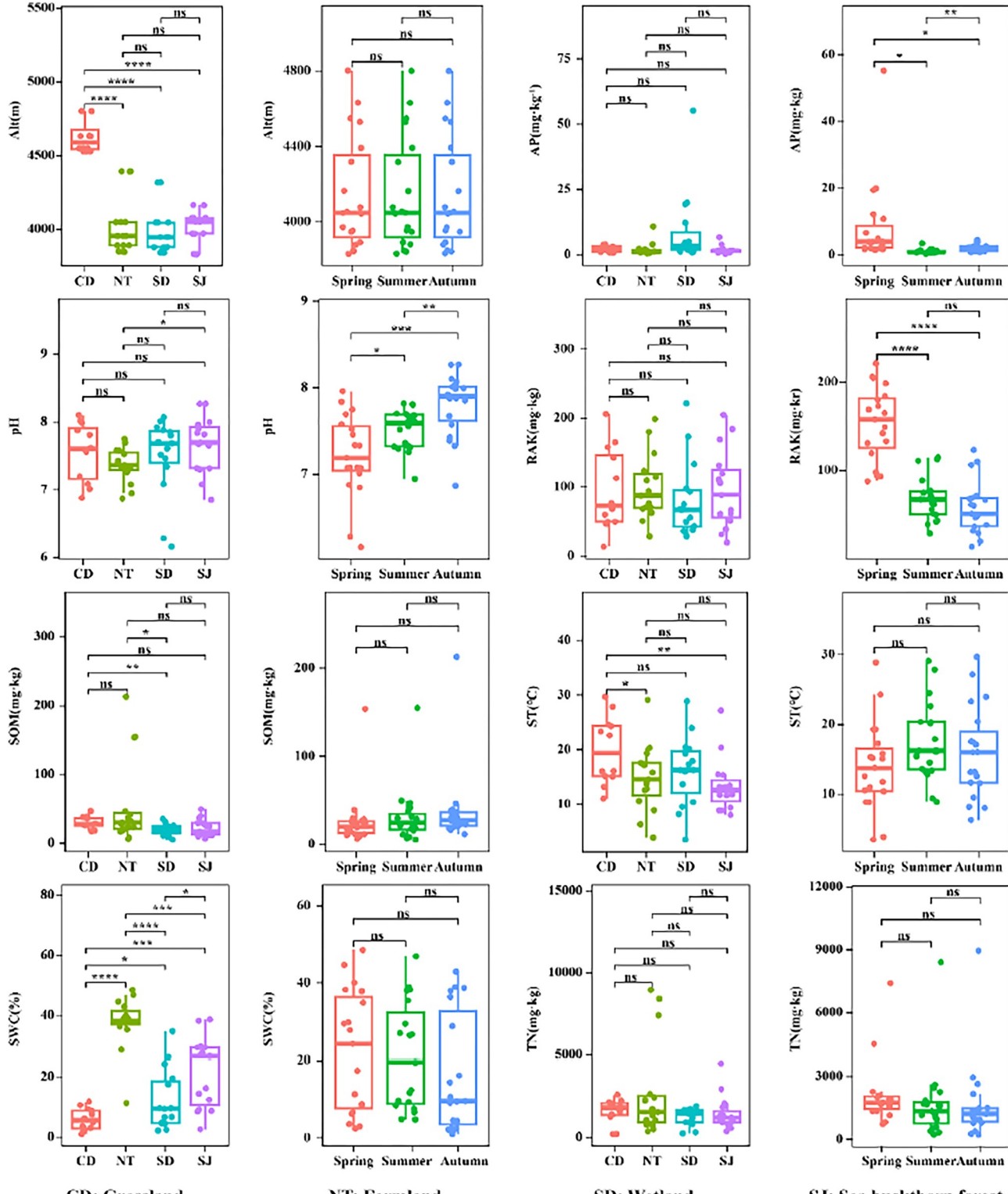

**Fig 6. Analysis of soil physicochemical factors in different spatiotemporal distribution in the Nianchu River Basin.**

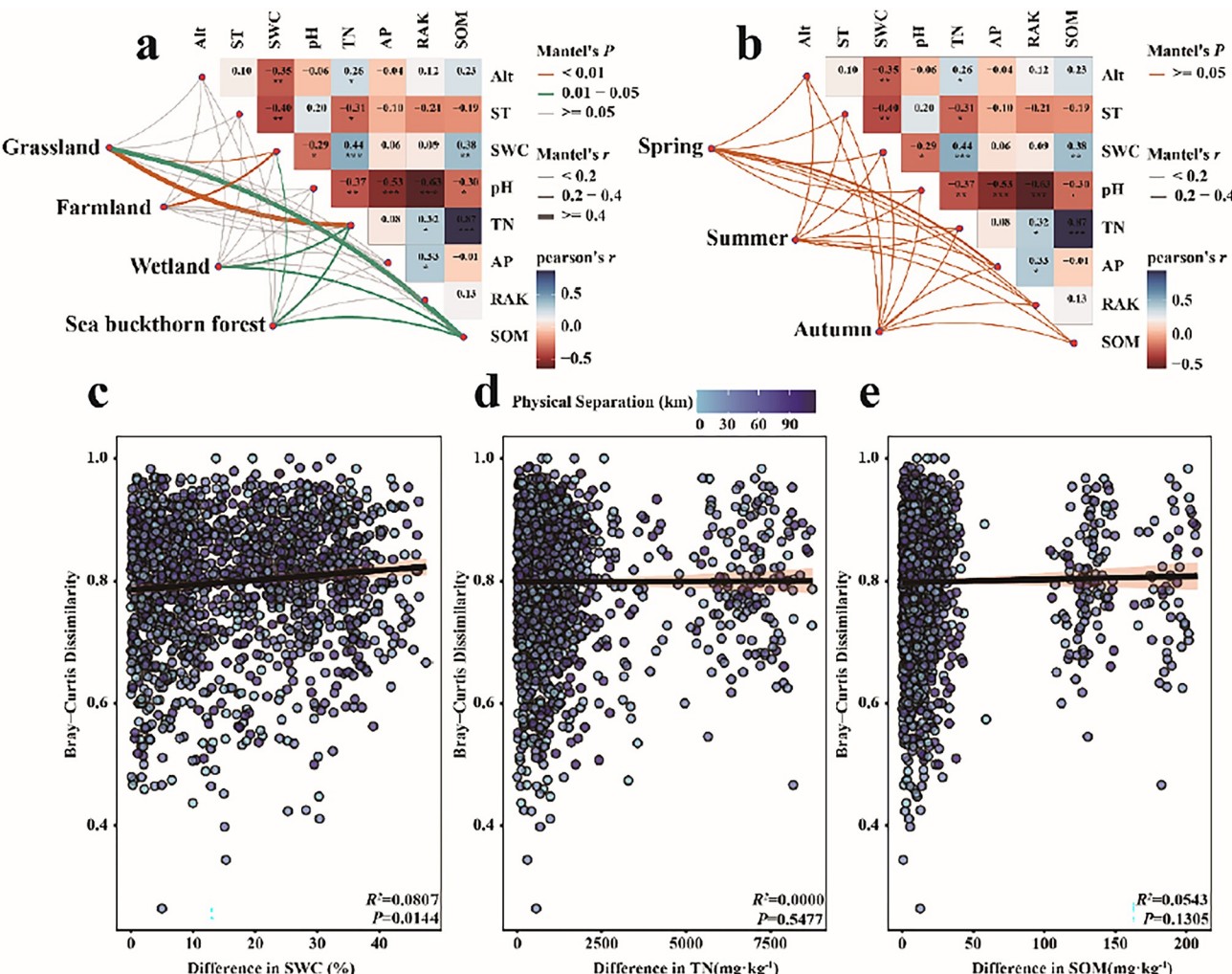

**Fig 7. Correlation between soil ciliates and environmental factors in different ecosystems and seasons.** a: Correlation between soil ciliates and environmental factors in different ecosystems; b: Correlation between soil ciliates and environmental factors in different seasons; c: Regression analysis of soil water content and soil ciliate communities; d: Regression analysis of total nitrogen and soil ciliate communities; e: Regression analysis of organic matter and soil ciliate communities.

stochastic diffusion was more strongly driven between ecosystems, and the stochastic diffusion process of soil ciliates was relatively weak among seasons.

## Discussion

### Community structure and its correlation with environmental factors

The non-submerged culture method, in-vivo observation and protargol staining were used to culture and identify soil ciliates at different spatial and temporal levels in the Nianchu River basin in order to reveal differences in their spatial and temporal distributions. A total of 199 species of soil ciliates were identified in the Nianchu River basin. Compared with the number of soil ciliate species in other studies [9,13], the number of species in this area is relatively large. It indicates that the soil ciliate species resources are abundant in the Qinghai-Tibet Plateau. Haptorida is the dominant group of soil ciliates in the Nianchu River Basin, followed by Prorodontida. These two groups dominate the soil ciliate communities and can obviously

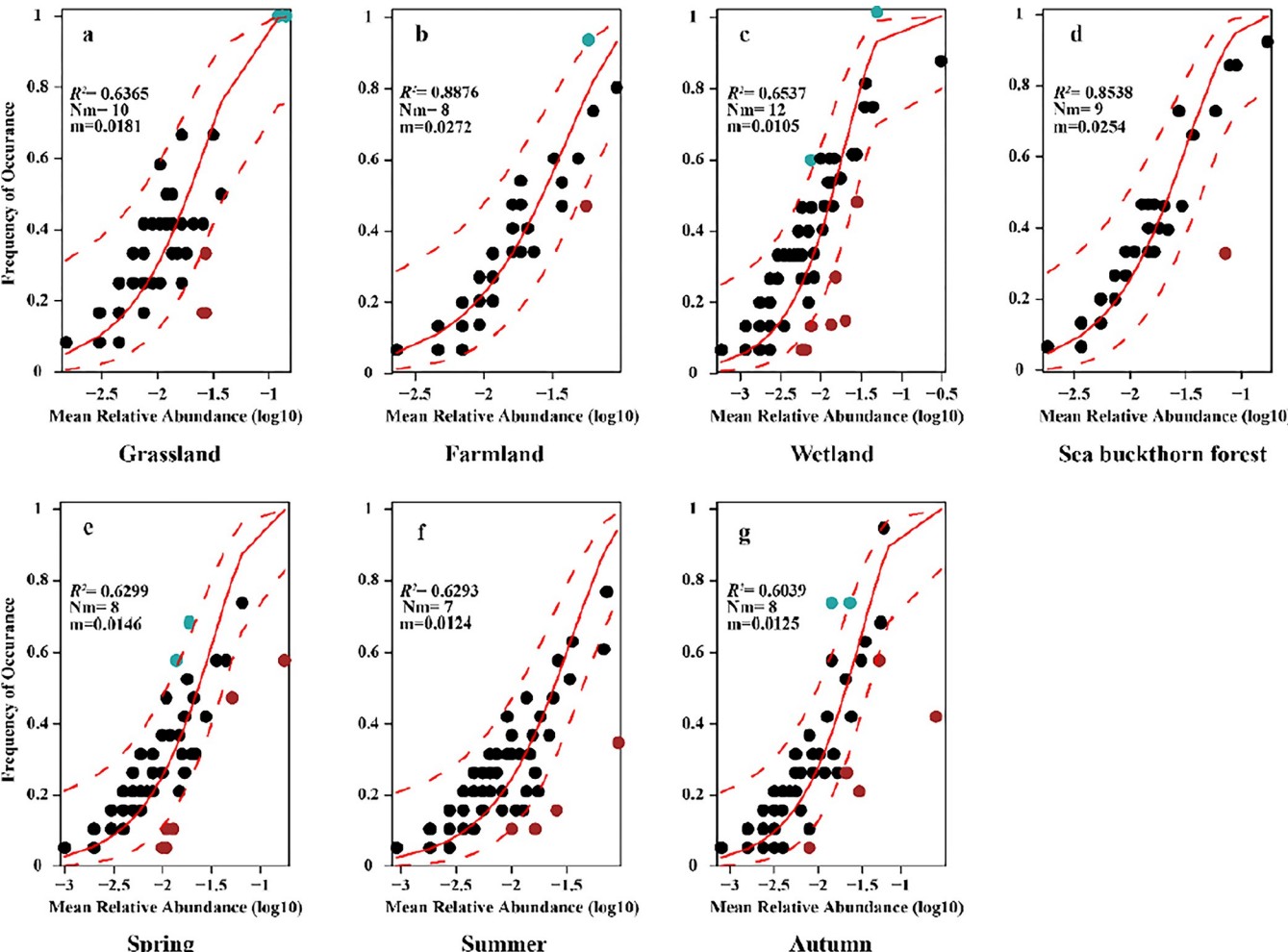

**Fig 8. Construction of soil ciliate community based on the neutral model.** a: Construction of soil ciliate community in grassland; b: Construction of soil ciliate community in farmland, c: Construction of soil ciliate community in wetland, d: Construction of soil ciliate community in sea buckthorn forest, e: Construction of soil ciliate community in spring, f: Construction of soil ciliate community in summer, g: Construction of soil ciliate community in autumn.

control the community structure [44,45]. The survival resources of other soil ciliate groups were preferentially occupied through competition, which affected the species composition of soil ciliate communities. In terms of temporal and spatial distribution, the dominant soil ciliates in Nianchu River Basin were mainly spirotricheans. The C/P quotient of soil ciliates was less than 1. This compares with a C/P quotient of 1.1 for ciliates in forests of Central Europe in 2005 [34]. The soil environment of Nianchu River basin is better than that of Central Europe. This is because spirotricheans tend to live in soils with better environmental (The change of soil environment is relatively stable). Colpodea tend to live in harsher soil environments (The change of soil environment is relatively drastic) [46]. The Nianchu River basin is located in the Qinghai-Tibet Plateau where the soil ecological environment is good, suitable for the growth and reproduction of soil ciliates. Colpodeans were not dominant in the Qinghai-Tibet Plateau soil environment. Grassland and farmland had the highest C/P quotient. This is because the grassland is located in the upper reaches of the Nianchu River basin, with high altitude, low temperatures, strong winds, and poor quality soil [2]. Farmland is cultivated in spring and harvested in autumn, and the soil environment is disturbed greatly. The C/P quotient of sea

buckthorn forest was the lowest of the four ecosystems. This is because the sea buckthorn forest is located in the valley area, is rarely visited, the soil organic matter content is high, and the soil environment is excellent. Soil ciliate cysts are present throughout the year and at the right time to decyst to regenerate soil ciliates. Therefore, the soil ciliates cultured in the laboratory can reflect the diversity of soil ciliate communities in the Nianchu River Basin, at least to some extent. Spatially, the highest value of the Shannon-Wiener diversity index was recorded in grassland, indicating that the soil ciliate community was the most complex of the four ecosystems investigated. Temporally, the lowest Shannon-Wiener diversity index was in autumn indicating that soil ciliate community was the least complex of the three seasons investigated. The richness index was the largest in the wetland. The lowest richness index in autumn. The results of non-metric scaling show that soil ciliate communities in the Nianchu River Basin have high similarity among seasons, while wetland soil ciliate communities have significant differences with other ecosystems. This is because the Nianchu River Basin is located in the Qinghai-Tibet Plateau, and the soil environment changes greatly between seasons. The low soil temperature in spring and autumn delays the greening period, and prolongs the yellowing period of the vegetation [47]. Only a few soil ciliates appear to be adapted to this soil environment, e.g., *Colpoda inflata*、 and *Halteria grandinella*. It is also the most suitable season for the survival of soil ciliates. However, soil ciliates exist in the form of cysts when the soil environment changes rapidly. When the soil environment is suitable for ciliate growth and reproduction, active ciliates will reappear in the soil. Soil samples from different sources made significant differences in soil ciliate communities between plots. But each season contains four types of plots. Some habitat-specific species that occur between plots may be found seasonally. This insignificant variation is seen in soil ciliate communities aggregated from all ecosystems. However, soil ciliate communities of individual ecosystem types may have significant differences between seasons. Ciliates living in water and living in soil have undergone a long period of adaptive evolution in their respective habitats, and the species composition and lifestyle of ciliates have undergone great changes. The SWC in the wetland of the Nianchu River Basin is high, while that in other ecosystems is relatively low. Therefore, wetland soil ciliate community has obvious difference with other ecosystems. A certain concentration of soil physical and chemical factors can promote the growth soil ciliates, and too high or too low concentration of soil physical and chemical factors maybe inhibit the growth and development of soil ciliates. The results of the Mantel test showed that SWC, TN and SOM were significantly correlated with the soil ciliate communities in the Nianchu River Basin. Further fitting showed that the SWC was the main environmental factor driving soil ciliate community structure (Fig 7C–7E), because the increase of SOM content could absorbed inorganic matter in soil into water-soluble organic compounds. Nitrogen is an indispensable element to construct the overall structure of soil ciliates, while SWC is a major limiting factor for the growth, reproduction and distribution of soil ciliates, therefore their life activities depend on the availability of soil water content [48].

## Co-occurrence network and keystone species

In soil ecosystems, soil ciliates often exist in complex networks, and co-occurrence networks can explain the potential interrelationships among species [49]. In co-occurrence network analyses, positive and negative correlations represent the synergistic and competitive relationships between species respectively. The spatial and temporal distributions of soil ciliates in the Nianchu River Basin were mainly positively correlated, indicating that the species of soil ciliate community is weak in species competition, and the species depend on each other through solidarity, cooperation or ecological complementarity, thus maintaining or enhancing the

diversity and stability of the community. Spatially, the number of nodes and edges of soil ciliate communities in sea buckthorn forest is low, indicating that the network structure was simple. Farmland has the largest clustering coefficient and mean path length in terms of time and space, because the soil ciliate biodiversity of the farmland ecosystem was relatively simple. Therefore, when disturbance occurs in the soil environment, the network will transmit the disturbance to the whole soil ciliate community in a short time. As a result, the structure of the whole network becomes unstable and the buffering ability of soil environment change is poor. Farmland also had the largest modularity coefficient, indicating that farmland soil ciliate species have more complex functions and more diversified ecological niches, because in the co-occurrence network, modularity is more likely to have independent functional niches [50]. However, long-term continuous and regular cultivation and fertilization in farmland resulted in more complex interrelationships and niche division among soil ciliate species. Keystone species play an important role in maintaining the structure of soil ciliate community network, and their disappearance will cause the collapse of the entire network [51]. Spatially, no keystone species were identified any of the four ecosystems in the Nianchu River Basin. Temporally, there was one keystone species in spring (*Spathidium muscorum*), one in autumn (*Cyclidium lanuginosum*), and two species in summer (*Ophryoglena atra* and *Prorodon virides*). These four keystone species play a key role in stabilizing network structure between seasons. Therefore, their disappearance and weakening may lead to fundamental changes in the entire soil ciliate community. These keystone species can be used as indicators to evaluate the diversity and dynamics of the soil ciliate community [52].

## Community assembly

Environmental screening is an important part of the deterministic process, and SWC significantly affected the formation of soil ciliate communities and network model in the Nianchu River Basin at the beginning of the year (Fig 7). Community assembly (The process by which species settle and interact to establish and sustain local communities through successive repeated migrations from regional species banks) based on neutral models showed that the construction of soil ciliate communities in the river basin is dominated by random processes. In terms of spatial and temporal distribution, the stochastic diffusion process was more strongly driven ecosystems than between seasons. Because random processes (birth, death, immigration, speciation and limited dispersal) are dominant in communities with high diversity and deterministic processes (interspecies interactions and environmental factors) are dominant in communities with low diversity [53], the inter-ecosystem soil ciliate diversity index in the Nianchu River Basin was higher than the inter-seasonal soil ciliate diversity index. Both ecosystem and seasonal m values were much less than 1, indicating that soil ciliate diffusion was strongly restricted, and that diffusion restriction drove the construction of soil ciliate communities in the Nianchu River Basin (Fig 8).

The α and β diversity (α diversity: the diversity of species in a habitat or community. β diversity: measures the rate of change of species composition from one community to another along a gradient at a regional scale) of soil ciliate communities in the Nianchu River Basin showed that these communities differed significantly in space, but not in time. Co-occurrence network analyses showed that the spatial and temporal distribution of soil ciliate communities was dominated by cooperative relationships and supplemented by competitive relationships. There were no ecosystem-based keystone species between habitats, and there were relatively low abundances of seasonal keystone species. SWC, TN and SOM were significantly correlated with soil ciliate communities in the Nianchu River Basin. Further analysis showed that SWC was the main environmental factor driving soil ciliate community structure.

## Supporting information

**S1 Fig. Species image data.**
(DOCX)

**S1 Table. Environmental factors.**
(XLSX)

**S2 Table. Abundance.**
(XLSX)

**S1 File.**
(DOCX)

## Acknowledgments

Thanks for the support of the high-level graduate program of Tibet University.

## Author Contributions

**Funding acquisition:** Bu Pu.

**Writing – original draft:** Shiying Zhu.

**Writing – review & editing:** Qian Huang, Tianshun Li, Mingyan Li, Qing Yang, Xiaodong Li, Alan Warren, Bu Pu.

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
