## [Decision Letter · Decision Letter 0]

8 Jan 2024

PONE-D-23-39236Soil water content drives spatiotemporal distribution and community assembly of soil ciliates in Nianchu River Basin, Qinghai-Tibet Plateau, ChinaPLOS ONE

Dear Dr. pu,

Thank you for submitting your manuscript to PLOS ONE. After careful consideration, we feel that it has merit but does not fully meet PLOS ONE’s publication criteria as it currently stands. Therefore, we invite you to submit a revised version of the manuscript that addresses the points raised during the review process.

** As managing editor, I have read the two reviews of your manuscript.  Due to the lack of detail, I have also closely read the manuscript and recommend that the manuscript be considered for publication pending extensive rewriting to make the data analysis more clear and provide a stronger foundation for the arguments made in the discussion.  Specifically, the article must be revised to make it more accessible the broader backgrounds of potential readers. **

**Since PLOS One is not a specialized ecology journal, definitions need to be provided for a broader audience. Examples- line 60 "community assembly", line 75 “driving factor and CONSTRUCTION MECHANISM’,  line 79 “sustainable development of biodiversity”, line 135 “neutral community model” (what does this mean and what alterative models were tested?), line 196 “co-occurence network” (how do you distinguish between correlations and functionally cooperative relationships), deterministic versus random models (Discussion- L361). Alpha and beta diversity need to be defined.**

**Materials and Methods were sufficiently detailed, and the reference to the Lynn classification of ciliate species was sufficient.  Since the classification of species in the different plots is central to the research, I agree with reviewer #2 that imaging data needs to be shown in supplemental data. An image of each species identified in the study should be provided for the different methods used to make your classification. The C/P quotient is defined sufficiently but I am not sure what you mean by r-selected and k-selected species).  It is not clear how the value obtained in samplings serves as an indicator of the overall health of the soil.**

**Starting with Table 3 (see also Table 4, Figure 3) data are provided for 4 groups designated CD, NT, SD and SJ. It is not clear from the main text, table or figure legends what these 4 designations are. I probably spent an hour trying to figure it out and never succeeded. The significance of the study cannot be assessed without a clear description of what is being analyzed. I don’t believe the findings for these groups were addressed in the Discussion.**

**Shannon and Margalef models: when I read the Materials and Methods, it seemed that both models are trying to assess similar phenomena.  However, the later model shows only one statistically significant difference (Margalef model) in panel C. Please elaborate more on what the two models are assessing and what the data tell you based on only one comparison yielding significant differences (and with only one method).**

**The work focuses on species identification (# of species) as a function of soil type (landscape) over time (3 seasons). Why isn’t any data  presented on the density of each species?  Are there statistically significant seasonal fluctuations in the ABUNDANCE of different species (akin to a summer bloom)? Are there any physical characteristics of the different  plot types that could be drivers of changes in community composition- temperature,  water content?**

**There are numerous comments in the discussion that are presented as factual findings, but that in my opinion are more sweeping generalizations that combine factual data with personal opinion.  Line 267- “soil ciliate resources are abundant”- what are these resources and why is higher biodiversity a marker for resource abundance? Lines 267-271: need to define what you mean by harsh and better soil quality. Lines 314-320 in Discussion on multiple physical parameters of soils seems to have contradictory statements in it. If not, I am just confused. Not sure if you are talking about biodiversity differences between plot types or lumping all ciliates together.  Community Assembly lines 352-368.  You need to define the term and better relate your data to the subject.  Are your conclusions rigorously drawn from the data or are you expressing an opinion that your data is consistent with.  Again- neutral model is discussed. Are there other models that you should consider?  **

We look forward to receiving your revised manuscript.

Kind regards,

Geoffrey M. Kapler

Academic Editor

PLOS ONE

Journal Requirements:

Whilst you may use any professional scientific editing service of your choice, PLOS has partnered with both American Journal Experts (AJE) and Editage to provide discounted services to PLOS authors. Both organizations have experience helping authors meet PLOS guidelines and can provide language editing, translation, manuscript formatting, and figure formatting to ensure your manuscript meets our submission guidelines. To take advantage of our partnership with AJE, visit the AJE website (http://aje.com/go/plos) for a 15% discount off AJE services. To take advantage of our partnership with Editage, visit the Editage website (www.editage.com) and enter referral code PLOSEDIT for a 15% discount off Editage services. If the PLOS editorial team finds any language issues in text that either AJE or Editage has edited, the service provider will re-edit the text for free.

5. Thank you for stating in your Funding Statement: 

 Comprehensive Scientific Investigation Project of Biodiversity Survey and Maintenance Mechanism Evaluation in the "One River and Four Rivers" Basin (Zangcai Science Education Index (2021) No.1 and Zangcai Education Index (2019) 01); Supported by "High-level Talents Cultivation Program" for Graduate Students of Tibet University, No.2020-GSP-S042.

7. We note that Figure 1 in your submission contain map images which may be copyrighted. All PLOS content is published under the Creative Commons Attribution License (CC BY 4.0), which means that the manuscript, images, and Supporting Information files will be freely available online, and any third party is permitted to access, download, copy, distribute, and use these materials in any way, even commercially, with proper attribution. For these reasons, we cannot publish previously copyrighted maps or satellite images created using proprietary data, such as Google software (Google Maps, Street View, and Earth). For more information, see our copyright guidelines: http://journals.plos.org/plosone/s/licenses-and-copyright.

We require you to either present written permission from the copyright holder to publish these figures specifically under the CC BY 4.0 license, or remove the figures from your submission:

Reviewers' comments:

Reviewer's Responses to Questions

**Comments to the Author**

1. Is the manuscript technically sound, and do the data support the conclusions?

Reviewer #1: Yes

Reviewer #2: No

2. Has the statistical analysis been performed appropriately and rigorously? 

Reviewer #1: Yes

Reviewer #2: I Don't Know

3. Have the authors made all data underlying the findings in their manuscript fully available?

Reviewer #1: Yes

Reviewer #2: No

4. Is the manuscript presented in an intelligible fashion and written in standard English?

Reviewer #1: Yes

Reviewer #2: No

5. Review Comments to the Author

Reviewer #1: Node calculations are not described in sufficient detail in the methods- how was inter-module connectivity (Zi) and inter-module connectivity (Pi) calculated?- also what is the difference between these?

Otherwise, descriptions of analyses done and conclusions reached are good, and appear to be original research.

Reviewer #2: The general study question is interesting, the diversity of ciliates in four different ecotypes and various seasons in

Nianchu River Basin, Qinghai-Tibet Plateau.

I'm sorry, but this ms is very very hard to read. There may be too many facts in the main text, some of which could be moved to supplementals. This might allow you to tell a story that would be compelling, either to a ciliate biologist, or to a quantitative ecologist. As it stands, I can't see either group being interested in struggling though the article. Think about the story you want to tell and direct your presentation toward that.

The methods that are put into quotes, such as ""five-point sampling method of plum blossom (Zhang et

103 al.,1991)"" are confusing, the quotes. And: this case was the second method I looked into, but can;t seem to find the sited ref.?

Your study rests on the identification of ciliate species via culturing, staining and then microscopy. Without at least some of this data (images) in the supplemental I have not way to evaluate the quality of this foundational data. This is also data that any ciliate biologist would want to look at.

6. PLOS authors have the option to publish the peer review history of their article (what does this mean?). If published, this will include your full peer review and any attached files.

Reviewer #1: No

Reviewer #2: No

---

## [Author Response · Author response to Decision Letter 0]

13 Feb 2024

1.Since PLOS One is not a specialized ecology journal, definitions need to be provided for a broader audience. Examples- line 60 "community assembly", line 75 “driving factor and CONSTRUCTION MECHANISM’, line 79 “sustainable development of biodiversity”, line 135 “neutral community model” (what does this mean and what alterative models were tested?), line 196 “co-occurence network” (how do you distinguish between correlations and functionally cooperative relationships), deterministic versus random models (Discussion- L361). Alpha and beta diversity need to be defined.

According to the reviewer’s comments, we have added explanations and changes to these contents. 

line 60, changed “community assembly” to “community assembly (The process by which species settle and interact to establish and sustain local communities through successive repeated migrations from regional species banks)”

line 75, changed “driving factors and construction mechanism” to “driving factors (factors affecting soil ciliate community) and community assembly mechanism”

line 79, changed “sustainable development of biodiversity” to “sustainable development of biodiversity (biodiversity is the basis, goal and means of sustainable development)”

line 135, we explain the neutral model as follows: Developed by sloan et al., it is widely used in microbial community analysis. It establishes the hypothesis that "randomness, birth, death, and migration play a key role in shaping the structure of these communities" by quantifying neutral processes. By fitting the relative abundence-frequency relationships of different groups observed in microbial studies to beta distributions derived from neutral theory. The model has been shown to be able to successfully replicate the importance of species abundance distribution observed in large biomes.

line 196, we make the following explanations and distinctions:

correlations：we use the values of positive correlation ratio and negative correlation ratio to express them.

functionally cooperative relationships: the positive correlation in functional cooperation is solidarity cooperation and co-evolution. The negative correlation in functional cooperation is competition and predation. This part of the content cannot be reflected in the collinear network, but can be further explained by the positive and negative correlation between the network nodes.

L361, “changed deterministic processes” to “deterministic processes (interspecies interaction and environmental factor)”

Changed “random processes” to “random processes (birth, death, immigration, speciation and limited dispersal) ‘

Alpha and beta diversity need to be defined, changed “α and β diversity” to “α and β diversity (α diversity: the diversity of species in a habitat or community. β diversity: measures the rate of change of species composition from one community to another along a gradient at a regional scale)”

2. Materials and Methods were sufficiently detailed, and the reference to the Lynn classification of ciliate species was sufficient. Since the classification of species in the different plots is central to the research, I agree with reviewer #2 that imaging data needs to be shown in supplemental data. An image of each species identified in the study should be provided for the different methods used to make your classification. The C/P quotient is defined sufficiently but I am not sure what you mean by r-selected and k-selected species). It is not clear how the value obtained in samplings serves as an indicator of the overall health of the soil.

According to the reviewer’s comments, we added the display imaging data of the species to upload in the form of supplementary material.

Our interpretation of r-selected and k-selected species is:

R- selected: choices that favor increasing the intrinsic growth rate are called R- selected. R- selected are pioneers of new habitats, more adaptable to the environment, but survival depends on chance, so they are "opportunistic" in a sense, prone to "sudden outbreaks and violent bankruptcies."

K- selected: choices that favor increased competitiveness are called K- selected. K- selected are defenders of a stable environment. In a certain sense, they are conservatives. When there is a disaster in the living environment, it is difficult to recover quickly, and if there are competitors to restrain them, they may tend to become extinct.

In the paper A huge, undescribed soil 489 ciliate (Protozoa: Ciliophpra) diversity in natural forest stands of Central Europe, Foissner W et al. gave a detailed description of the value obtained by C/P as an indicator of the overall health status of soil.

When C⁄P≤1, the environment in the soil habitat is relatively good. When C⁄P＞1, the environment in the soil habitat is relatively bad.

3. Starting with Table 3 (see also Table 4, Figure 3) data are provided for 4 groups designated CD, NT, SD and SJ. It is not clear from the main text, table or figure legends what these 4 designations are. I probably spent an hour trying to figure it out and never succeeded. The significance of the study cannot be assessed without a clear description of what is being analyzed. I don’t believe the findings for these groups were addressed in the Discussion.

According to the reviewer’s comments, we made changes to the pictures in the manuscript.

Changed “CD” to “Grassland”, changed “NT” to “Farmland”, changed “SD” to “Wetland”, changed “SJ” to Sea buckthorn forest”

4. Shannon and Margalef models: when I read the Materials and Methods, it seemed that both models are trying to assess similar phenomena. However, the later model shows only one statistically significant difference (Margalef model) in panel C. Please elaborate more on what the two models are assessing and what the data tell you based on only one comparison yielding significant differences (and with only one method).

According to the reviewer’s comments, we make the following reply.

Margalef index reflects community species richness：The abundance of species in a community or environment, and an index of species richness in a biome (or sample). The larger the index, the higher the species richness in the ecosystem. However, this index has some limitations and needs to be combined with other diversity indexes. The index takes into account only the number and individual abundance of species, not the function and niche of species. Its application can help us understand the stability of ecosystems, environmental change and ecological restoration effects.

Shannon-Wiener index reflects community species diversity based on the number of species. The index is concerned not only with species richness, but also with species evenness, so it is a more comprehensive response to community structure.

Based on data from a comparison that produces a significant difference, the following conclusions can be drawn: Shannon-Weiner diversity index of soil ciliate community was significantly different in different time and space. These results indicate that the distribution of soil ciliate community is not uniform in different time and space, and the complexity of the community is different.

The Margalef index of soil ciliate community had significant differences among plots, but no significant differences in seasons. These results indicated that the species richness of soil ciliate community was different between plots, and the species richness of soil ciliate community was the same in season.

5. The work focuses on species identification (# of species) as a function of soil type (landscape) over time (3 seasons). Why isn’t any data presented on the density of each species? Are there statistically significant seasonal fluctuations in the ABUNDANCE of different species (akin to a summer bloom)? Are there any physical characteristics of the different plot types that could be drivers of changes in community composition- temperature, water content?

According to the reviewer’s comments, we uploaded the species density data to the database and the resulting link is as follows: DOI:10.6084/m9.figshare.25183643

There are no statistically significant seasonal fluctuations in the abundance of different species.

We conducted one-way ANOVA for different types of environmental factors and found that (Fig 6): Elevation, soil organic matter, soil temperature and soil water content were significantly different among different plots, suggesting that they might be the driving factors of community composition change. However, through mantel test and further analysis of these environmental factors, we found that soil water content was the main environmental factor driving soil ciliate community change.

6. There are numerous comments in the discussion that are presented as factual findings, but that in my opinion are more sweeping generalizations that combine factual data with personal opinion. Line 267- “soil ciliate resources are abundant”- what are these resources and why is higher biodiversity a marker for resource abundance? Lines 267-271: need to define what you mean by harsh and better soil quality. Lines 314-320 in Discussion on multiple physical parameters of soils seems to have contradictory statements in it. If not, I am just confused. Not sure if you are talking about biodiversity differences between plot types or lumping all ciliates together. Community Assembly lines 352-368. You need to define the term and better relate your data to the subject. Are your conclusions rigorously drawn from the data or are you expressing an opinion that your data is consistent with. Again- neutral model is discussed. Are there other models that you should consider?

According to the reviewer’s comments,

Line 267 We reviewed the description of the article and found that our description was inaccurate. Changed “Compared with other soil ciliate diversity studies[9,13], the number of soil ciliate species in this area is relatively large. These results indicate that soil ciliate resources are abundant in the Qinghai-Tibet Plateau.” to “Compared with the number of soil ciliate species in other studies[9,13], the number of species in this area is relatively large. It indicates that the soil ciliate species resources are abundant in the Qinghai-Tibet Plateau.”

Lines 267-271. We explain harsh and better soil quality. Changed “This is because soil ciliates of Spirotrichea tend to live in soils with better soil environmental quality. Soil ciliates of Colpodea tend to live in harsher soil environments” to “This is because soil ciliates of Spirotrichea tend to live in soils with better soil environmental (The change of soil environment is relatively stable). Soil ciliates of Colpodea tend to live in harsher soil environments (The change of soil environment is relatively drastic)”

Lines 314-320 We explain the content of this part. In the discussion of soil physical and chemical factors, we concentrated all soil ciliates and conducted an overall analysis of the environmental factors affecting soil ciliates. Because the screened soil water content, total nitrogen and organic matter are correlated with different plots.

lines 352-368 We are redefining the terms for the content of Community Assembly. Changed “community assembly” to “community assembly (The process by which species settle and interact to establish and sustain local communities through successive repeated migrations from regional species banks)”

Changed “deterministic processes” to “deterministic processes (interspecies interactions and environmental factors)”

Change “random processes” to “random processes (birth, death, immigration, speciation and limited dispersal)”

Our conclusions are drawn strictly from the data, and the views expressed are consistent with our data.

The neutral community model (NCM) quantifies the importance of stochastic processes.

R2 represents the overall fit of the neutral community model.R2 The higher the R2 value, the closer to the neutral model, it means that the community construction is more affected by random processes and less affected by deterministic processes.

N describes the metacommunity size, which in the text is the total abundance of all species in each sample.

m quantifies mobility at the community level, and the smaller the m value, the more restricted the dispersal of species throughout the community.

Therefore, the neutral model is sufficient and no other model is needed.

7.Node calculations are not described in sufficient detail in the methods- how was inter-module connectivity (Zi) and inter-module connectivity (Pi) calculated?- also what is the difference between these?

According to the editor's comments, we have added the calculation formulas for Zi and Pi to the manuscript. Zi is the z-scores of the connectivity degree within modules of node i, and Pi is the participation coefficient calculated according to the connectivity degree between modules of node i.

Z_i=(K_i-(K_si)®)/〖σK〗_si ⑴

P_i=1-∑_(s=1)^(N_m)▒(K_si/K_i )^2 ⑵

Where, Zi is the connectivity degree of node i module, Ki indicates the connectivity of module i of node, (K_si)® is the average value of Ki of all nodes in module s where node i resides, 〖σK〗_si is the standard deviation of connectivity within the module of all nodes in module s where node i resides, Pi is the participation coefficient of node i, Nm is the number of modular, s indicates the module s, and Kis is the connectivity of node i in each module

According to the editor's comments, we have made changes to the format of the manuscript.

9. Did you know that depositing data in a repository is associated with up to a 25% citation advantage (https://doi.org/10.1371/journal.pone.0230416)? If you’ve not already done so, consider depositing your raw data in a repository to ensure your work is read, appreciated and cited by the largest possible audience. You’ll also earn an Accessible Data icon on your published paper if you deposit your data in any participating repository (https://plos.org/open-science/open-data/#accessible-data).

According to the editor's comments, we store the data in a repository, forming a link as: DOI: 10.6084/m9.figshare.25183643

10. We suggest you thoroughly copyedit your manuscript for language usage, spelling, and grammar. If you do not know anyone who can help you do this, you may wish to consider employing a professional scientific editing service.

According to the editor's comments, We invited Dr. Alan Warren from the Natural History Museum to make changes to the language of the manuscript. And add it to the author sequence as a collaborator. This was agreed upon by all the authors. Alan Warren is an expert in the systematics and biodiversity of ciliated protists (ciliates). We have sent a request email to the editorial office to add Dr. Alan Warren to the author sequence.

11. We note that the grant information you provided in the ‘Funding Information’ and ‘Financial Disclosure’ sections do not match. When you resubmit, please ensure that you provide the correct grant numbers for the awards you received for your study in the ‘Funding Information’ section.

According to the editor's comments, We have resubmitted the project number of the scholarship.

Funding: Comprehensive Scientific Investigation Project of Biodiversity Survey and Maintenance Mechanism Evaluation in the "One River and Four Rivers" Basin (Zangcai Science Education Index (2021) No.1 and Zangcai Education Index (2019) 01); Supported by "High-level Talents Cultivation Program" for Graduate Students of Tibet University, No.2020-GSP-S042. There was no additional external funding received for this study.

12. Please provide an amended statement that declares all the funding or sources of support (whether external or internal to your organization) received during this study, as detailed online in our guide for authors at http://journals.plos.org/plosone/s/submit-now. Please also include the statement “There was no additional external fundi

---

## [Editor Report · Decision Letter 1]

16 Feb 2024

Soil water content drives spatiotemporal the distribution and community assembly of soil ciliates in the Nianchu River Basin, Qinghai-Tibet Plateau, China

PONE-D-23-39236R1

Dear Dr. Pubu,

We’re pleased to inform you that your manuscript has been judged scientifically suitable for publication and will be formally accepted for publication once it meets all outstanding technical requirements.

Kind regards,

Geoffrey M. Kapler

Academic Editor

PLOS ONE

Additional Editor Comments (optional):

The revised manuscript is vastly improved with regard to clarity of the text descriptions of general concepts in the scope of ecological studies, data analysis and conclusions drawn from the data. The revised manuscript should be of general interest to field biologists in general and ciliate molecular biologists. Hence, I am of the opinion that this work should be accepted for publication. The supplementary data files- images of the species identified in this study- are an important addition to the data presented.

Minor changes needed for provisional acceptance:

Figure 1: error- CD was designated for two areas- grasslands and wetlands. Change to SD for wetlands.

Figure 2C and D legends: the text is wrong since it describes Venn diagrams. The data and legend are not aligned.

p 70,line 277 missing information on cited work- "CP quotient of XX for ciliates in forest of Central Europe'

Typographical errors:

p 55 line 1: change to 'the spatiotemporal'

p 59 line 105: change to 'quadrants'

p 55, line 107: change form to 'from'

p 64: 'for the four ecosystems'

p 70, line 313: spelling error 'temporally'

p 74: duplicate word "low low"

Change header DISCUSS to DISCUSSION.

Eliminate CONCLUSION header- you draw conclusions in the discussion.

Obtain input from PLOS One copy editor on how to display table figure legend titles and text. They need to be cleaned up for publication. (i.e. Tab2.Composition....; Fig1. This is the Fig1 title.This is the Fig 1 legend.

MOve tables out of the body of the text and append them to the end of the text as described in Instructions to Authors.
---

## [Editor Report · Acceptance letter]

1 Jul 2024

PONE-D-23-39236R1 

PLOS ONE

Dear Dr. pu, 

I'm pleased to inform you that your manuscript has been deemed suitable for publication in PLOS ONE. Congratulations! Your manuscript is now being handed over to our production team.

Kind regards, 

on behalf of

Dr. Geoffrey M. Kapler 

Academic Editor

PLOS ONE